# Giant Faraday rotation in atomically thin semiconductors

Benjamin Carey [1,2], Nils Kolja Wessling[1,3], Paul Steeger[1], Robert Schmidt [1], Steffen Michaelis de Vasconcellos [1], Rudolf Bratschitsch[1] ✉ & Ashish Arora[1,4] ✉

Faraday rotation is a fundamental effect in the magneto-optical response of solids, liquids and gases. Materials with a large Verdet constant find applications in optical modulators, sensors and non-reciprocal devices, such as optical isolators. Here, we demonstrate that the plane of polarization of light exhibits a giant Faraday rotation of several degrees around the A exciton transition in hBN-encapsulated monolayers of $WSe_2$ and $MoSe_2$ under moderate magnetic fields. This results in the highest known Verdet constant of $-1.9 \times 10^7 \deg T^{-1} cm^{-1}$ for any material in the visible regime. Additionally, interlayer excitons in hBN-encapsulated bilayer $MoS_2$ exhibit a large Verdet constant ($V_{IL} \approx +2 \times 10^5 \deg T^{-1} cm^{-2}$) of opposite sign compared to A excitons in monolayers. The giant Faraday rotation is due to the giant oscillator strength and high $g$-factor of the excitons in atomically thin semiconducting transition metal dichalcogenides. We deduce the complete in-plane complex dielectric tensor of hBN-encapsulated $WSe_2$ and $MoSe_2$ monolayers, which is vital for the prediction of Kerr, Faraday and magneto-circular dichroism spectra of 2D heterostructures. Our results pose a crucial advance in the potential usage of two-dimensional materials in ultrathin optical polarization devices.

The Faraday effect is crucial for numerous scientific and technological advancements in astronomy, biology, chemistry, physics, and materials science. For instance, it is used for investigating the magnetic domain structure in solids[1,2], nuclear magnetic resonance in fluids via optical detection[3,4], paramagnetic gas molecule detection[5], determination of magnetic fields[6] and electron-density distribution in outer space and celestial objects[7], probing spin coherence in cold atoms[8], quantum spin fluctuation measurements[9], biochemical and biomolecular detection[10], stabilization of laser frequency[11], optical current sensing[12], optical Hall effect[13], and optical isolators[14].

In recent years, extraordinary progress has been made in exploring the unique physical phenomena in atomically thin transition metal dichalcogenide (TMDC) semiconductors[15–21]. In two-dimensional TMDCs, Coulomb-bound electron–hole composite quasiparticles such as neutral and charged excitons possess large binding energies and giant oscillator strengths[19,21], when compared to the traditional group III–V or II–VI quantum wells[22,23]. These exceptional properties enable investigating 2D quantum effects even at room temperature. Furthermore, the corners of the Brillouin zone i.e. the $K^{\pm}$ valleys in TMDCs selectively couple to circularly polarized light[21,24]. The magnetic moments associated with the neighboring $K^+$ and $K^-$ valleys are opposite to each other. This property leads to effects unique to atomically thin TMDCs, such as valley polarization[24], valley coherence[25], valley Zeeman splitting[26,27], the valley Hall effect[28], the valley-selective optical Stark effect[29], and magnetic-field-induced valley polarization[30–37]. From the magneto-optics perspective, TMDCs have been extensively studied using magneto-photoluminescence, magneto-reflectance and magneto-transmittance[21]. However, the

[1]Institute of Physics and Center for Nanotechnology, University of Münster, Wilhelm-Klemm-Strasse 10, Münster, Germany. [2]School of Mathematics and Physics, The University of Queensland, St Lucia, QLD, Australia. [3]Institute of Photonics, Department of Physics, University of Strathclyde, 99 George Street, Glasgow, UK. [4]Department of Physics, Indian Institute of Science Education and Research, Dr. Homi Bhabha Road, Pune, Maharashtra, India. ✉e-mail: rudolf.bratschitsch@uni-muenster.de; ashish.arora@iiserpune.ac.in

classic magneto-optical phenomena such as the Faraday and Kerr effect still remain to be explored experimentally due to major challenges involved in these measurements on the microscopic level[38–40].

For 2D materials, the Faraday and Kerr effects can provide crucial information on the valley-related processes, such as valley Zeeman splitting[38] and magnetic-field-induced valley polarization[38]. While Faraday rotation is generally accompanied with ellipticity when light passes through an absorbing medium (Fig. 1), a measurement of only Faraday rotation is sufficient for obtaining all the magneto-optical information about the sample[1].

Here, we show that the giant oscillator strength and large valley Zeeman splitting of excitons in WSe$_2$ and MoSe$_2$ monolayers result in the highest known Verdet constant (Faraday rotation per unit length per unit magnetic field) in a material in the visible regime[31–36,41–48]. We measure giant Faraday rotations on the order of degrees under a small magnetic field of around 1 T.

Faraday rotation arises from the non-zero off-diagonal term of the complex dielectric tensor of a material. For light incident along the $z$-direction perpendicular to the sample plane with $B\|z$, (i.e. Faraday geometry), the complex dielectric tensor for the $xy$ plane can be written as[1,49–51]

$$\overset{\leftrightarrow}{\tilde{\epsilon}} = \begin{pmatrix} \tilde{\epsilon}_{xx} & \tilde{\epsilon}_{xy} \\ -\tilde{\epsilon}_{xy} & \tilde{\epsilon}_{yy} \end{pmatrix} \tag{1}$$

The diagonal components are responsible for the conventional optical response of materials in reflectance, transmittance, and absorption. For materials with a high degree of in-plane symmetry, such as monolayer $MX_2$ ($M$ = Mo, W; $X$ = S, Se), the diagonal components are equal i.e. $\tilde{\epsilon}_{xx} = \tilde{\epsilon}_{yy}$. Normally, in semiconductors, off-diagonal terms $\tilde{\epsilon}_{xy}$ are zero in the absence of a magnetic field[49]. But these components reach very large values around the exciton resonances under magnetic fields, due to the exciton Zeeman splitting[39]. We note that in two-dimensional electron/hole gases in the quantum Hall regime, $\tilde{\epsilon}_{xy}$ also rises sharply around inter-Landau level transition energies[52]. However, usually, the magnetic field to reach the quantum Hall regime is much higher compared to the one required for large Faraday effects around Zeeman-split exciton energies[1,53]. Far from the exciton lines, $\tilde{\epsilon}_{xy}$ are smaller by many orders of magnitude. $\tilde{\epsilon}_{xy}$ is given as[49,54]

$$\tilde{\epsilon}_{xy}(B) = \epsilon_{xy1} + i\epsilon_{xy2} = \frac{\tilde{V}\lambda_0 \tilde{n}_{xx} B}{\pi} \tag{2}$$

where $\tilde{n}_{xx} = n + ik = \sqrt{\tilde{\epsilon}_{xx}} = \sqrt{\epsilon_{xx1} + i\epsilon_{xx2}}$ is the complex refractive index, and $\tilde{V} = V_{FR} + iV_{FE}$ is the complex Verdet constant with its components $V_{FR}$ and $V_{FE}$. The complex Verdet constant is related to the complex Faraday rotation $\tilde{\phi}_F = \phi_F + i\eta_F$ as $\tilde{V} = \tilde{\phi}_F/(d\,B)$ where $\phi_F$ and $\eta_F$ are Faraday rotation and Faraday ellipticity, respectively, $d$ is the sample thickness, $B$ is the magnetic field. We note that $\tilde{\epsilon}_{xy}$ is related to the popularly defined "Voigt constant" $Q$ as $Q = i\tilde{\epsilon}_{xy}/\tilde{\epsilon}_{xx}$[50].

## Results

### Excitonic Faraday rotation in MoSe$_2$ and WSe$_2$ monolayers

We measure the Faraday rotation (the real component of $\tilde{\phi}_F$, i.e., $\phi_F$) of light around the neutral and charged A exciton transitions in a hBN-encapsulated MoSe$_2$ monolayer, and the neutral A exciton transition in a hBN-encapsulated WSe$_2$ monolayer. The substrates are c-cut double-side polished sapphire, which enable optical transmission measurements. Details about our experimental setup are described in ref. 38 and are briefly summarized in the supporting information. Here, we first discuss the case of the MoSe$_2$ sample, and extend our conclusions to the WSe$_2$ sample afterwards. Figure 2a shows the measured optical transmittance spectrum of an hBN-encapsulated MoSe$_2$ monolayer in the spectral region of the neutral and charged exciton (trion) at a temperature of $T$ = 10 K. The measured spectrum (solid spheres) is modeled using the transfer-matrix method (lines) to incorporate the effects of optical interference due to multiple layers of the sample on the spectral line shape[55,56]. The excitonic contribution to the dielectric function is described as a complex Lorentzian

$$\tilde{\epsilon}_{xx}(E) = \epsilon_{xx1} + i\epsilon_{xx2} = (n_b + ik_b)^2 + \sum_j \frac{A_j}{E_{0j}^2 - E^2 - i\gamma_j E} \tag{3}$$

where, $A_j$, $E_{0j}$, and $\gamma_j$ are the oscillator strength, transition energy and full-width at half-maximum (FWHM) linewidth of the $j^{th}$ resonance. $n_b + ik_b$ is the complex background dielectric function of monolayer MoSe$_2$ without excitonic contributions[57]. The exciton and trion resonances A and T are at 1.634 eV and 1.607 eV, respectively (Fig. 2a). Narrow FWHM linewidths (2.7 meV and 4.0 meV, respectively) approaching the homogeneous linewidth limit indicate the excellent quality of our samples[58–60].

Faraday rotation $\phi_F$ is measured in this spectral region under moderate out-of-plane applied magnetic fields ranging from $B$ = 0.2–1.4 T. We observe characteristic Faraday rotation line shapes with a dip around the exciton and trion energies. From our data, we determine the real and imaginary components ($\epsilon_{xy1}$ and $\epsilon_{xy2}$) of the off-diagonal dielectric constant $\tilde{\epsilon}_{xy}$ of monolayer MoSe$_2$ following Eq. (2). An example of $\epsilon_{xy1}$ and $\epsilon_{xy2}$ for $B$ = 1.4 T is shown in Fig. 2c. As required by Eq. (2), the procedure involves a calculation of Faraday ellipticity using a Kramers–Kronig analysis of our data in Fig. 2b[61], as well as the complex diagonal dielectric function $\tilde{\epsilon}_{xx}$. Details of the calculation are provided in the supporting information.

The Faraday rotation line shapes in Fig. 2b are modeled using the transfer-matrix method to determine the Zeeman splittings of A and T as a function of magnetic field (Fig. 2d)[38,55,56]. First, the transmittance spectrum is modeled as described before. The complex transmission Fresnel coefficients for left and right circular polarizations $\sigma^\pm$ are given as

$$\tilde{t}_\pm = t_\pm e^{i\phi_\pm} \tag{4}$$

They are identical in the absence of an external magnetic field. $\tilde{t}_\pm$ are obtained from modeling the transmission spectrum. In the presence of a magnetic field, excitons and trions undergo a valley Zeeman splitting, leading to different $\tilde{t}_\pm$, and thereby different $\phi_\pm$. The Faraday rotation of polarized light under a magnetic field is related to $\phi_\pm$ as

$$\phi_F = -\frac{1}{2}(\phi_+ - \phi_-) \tag{5}$$

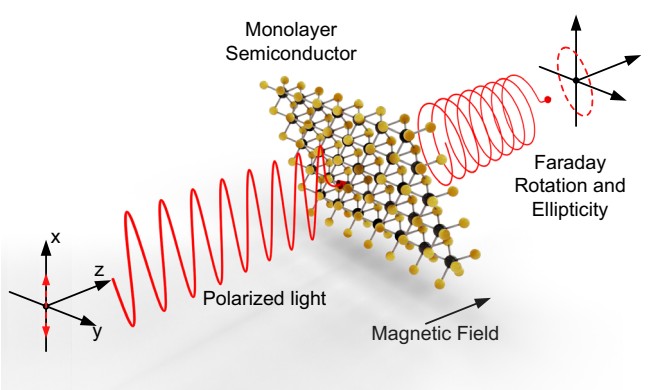

**Fig. 1 | Faraday effect in a 2D semiconductor.** Schematic drawing depicting how linearly polarized light passes through an atomically thin semiconductor under a magnetic field and acquires Faraday rotation and ellipticity.

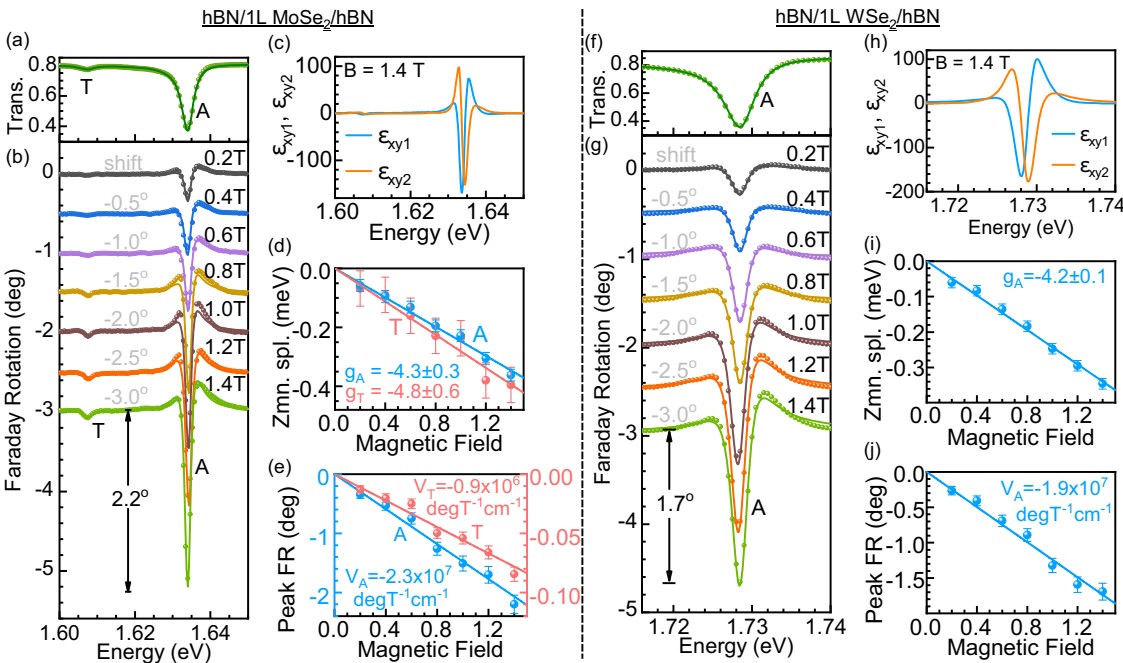

**Fig. 2 | Excitonic Faraday rotation in hBN-encapsulated monolayers of MoSe₂ and WSe₂. a** Optical transmission spectrum of hBN/1L MoSe₂/hBN (left panel) and **f** hBN/1L WSe₂/hBN (right panel) around the exciton and trion resonances A and T in the MoSe₂ sample and A exciton in the WSe₂ sample. A strong dip is observed around the neutral A exciton resonance. **b, g** Experimental (spheres) and modeled (solid lines) Faraday rotation spectra of resonances in magnetic fields ranging from $B = 0.2\,T – 1.4\,T$ for the two samples. The spectra are vertically shifted by 0.5° successively for clarity. The shift is mentioned along with the respective plots. Characteristic Faraday rotation spectral line shapes are revealed around the resonance energies. The exciton transition exhibits a large peak Faraday rotation of 2.2°

and 1.7°, respectively, for the MoSe₂ and WSe₂ cases at $B = 1.4\,T$. **c, h** Real and imaginary parts of the off-diagonal dielectric function of the two materials at $B = 1.4\,T$ derived from our measurements, as explained in the main text and the supporting information. **d, i** Zeeman splittings of the resonances derived from line shape modeling. Linear fits (solid lines) are used to derive the effective g-factors $g_A$ and $g_T$. **e, j** Measured peak Faraday rotation of the resonances as a function of the magnetic field. Peak rotation is fitted linearly for deriving the Verdet constants $V_A = -(2.3 \pm 0.2) \times 10^7 \, deg \, T^{-1} \, cm^{-1}$ and $V_T = -(0.9 \pm 0.2) \times 10^6 \, deg \, T^{-1} \, cm^{-1}$ for the MoSe₂ and $V_A = -(1.9 \pm 0.2) \times 10^7 \, deg \, T^{-1} \, cm^{-1}$ for the WSe₂ sample.

Solid lines in Fig. 2b are the fits to the Faraday rotation spectra using this procedure. The modeling provides the valley Zeeman splittings of the excitons and trions as a function of magnetic field, which are plotted in Fig. 2d. The absolute value of the Zeeman splitting increases linearly as a function of the magnetic field. The Zeeman splitting of an exciton resonance X is given as $\Delta E = g_X \mu_B B$, where $g_X$ is the effective exciton g-factor and $\mu_B = 5.788 \times 10^{-5} \, eV \, T^{-1}$ is the Bohr's magneton. By fitting the data in Fig. 2d using this relation, we obtain the exciton and trion g-factors as $g_A = -4.3 \pm 0.3$ and $g_T = -4.8 \pm 0.6$. These values are in excellent agreement with literature values[21,33,45,46]. Figure 2f–j presents the results for an hBN-encapsulated WSe₂ monolayer around the A exciton, yielding similar results as for MoSe₂.

We find that around the A exciton, the magnitude of the Faraday rotation relative to the flat spectral background (i.e., 0° rotation) is 2.2° for monolayer MoSe₂, and 1.7° for monolayer WSe₂ at $B = 1.4\,T$. The magnitude of Faraday rotation for the A and T transitions in MoSe₂ are plotted as a function of the magnetic field in Fig. 2e, and for the A exciton in WSe₂ in Fig. 2j. The rotation increases linearly with the magnetic field for all resonances. The Faraday rotation (real part) around a resonance X at a magnetic field $B$ is given as

$$\phi_F = V_X \, d \, B \qquad (6)$$

where $V_X$ is the Verdet constant of the resonance, and $d$ is the layer thickness. Fitting the data in Figs. 2e, j using this relation yields Verdet constants of A and T, respectively as $V_A = -(2.3 \pm 0.2) \times 10^7 \, deg \, T^{-1} \, cm^{-1}$ and $V_T = -(0.9 \pm 0.2) \times 10^6 \, deg \, T^{-1} \, cm^{-1}$ for monolayer MoSe₂, and $V_A = -(1.9 \pm 0.2) \times 10^7 \, deg \, T^{-1} \, cm^{-1}$ for monolayer WSe₂.

## Faraday rotation around interlayer excitons in a MoS₂ bilayer

As an example of a material with a positive Verdet constant, we perform Faraday rotation spectroscopy of interlayer excitons (IL) in a hBN-encapsulated MoS₂ bilayer (Fig. 3). Interlayer excitons in bilayer and bulk TMDCs are known to have a positive g-factor which is opposite in sign to the intralayer excitons[21,62,63]. However, interlayer excitons have a much smaller oscillator strength and a larger line width compared to the intralayer excitons[62–64]. Therefore, their Verdet constant is expected to be smaller. The measured transmittance (for $B = 0$) and the Faraday rotation spectra (under $B = 0.4 – 1.4\,T$) of the hBN-encapsulated MoS₂ bilayer sample are shown in Fig. 3a. Clear signatures corresponding to the intralayer exciton (A) at 1.930 eV, intralayer trion (T) at 1.909 eV, and a split interlayer exciton (IL₁ at 1.994 eV and IL₂ at 2.004 eV) are visible in the Faraday rotation spectra. The assignment of the features are performed on the following grounds: A and T resonances have similar g-factors, nearly equal to −4, suggesting their intralayer character[21,63]. T polarizes strongly under magnetic field, with its valley polarization approaching ≈ +14% (Fig. 3c). Furthermore, A polarizes only weakly (valley polarization ≈ −2%). The large polarization of T with an opposite sign compared to A is characteristic for the appearance of a trion–exciton pair[65]. We notice that the binding energy of the trion in the MoS₂ bilayer is about 21 meV. In comparison, the reported value in a non-encapsulated bilayer is 27 meV[66]. A smaller value in our work signifies the effect of an increased dielectric constant around the trion, due to hBN encapsulation. The split IL exciton lines are identified due to their positive g-factors ($g_{IL1} = +6.6 \pm 0.3$ and $g_{IL2} = +7.2 \pm 0.3$)[21,62,63] in Fig. 3d. In previous works, one IL resonance has been observed in optical reflectance spectra[21,63]. In our transmittance spectra, we also notice one (broad) IL line (Fig. 3a), while Faraday rotation spectroscopy is able to resolve two close-lying IL features due

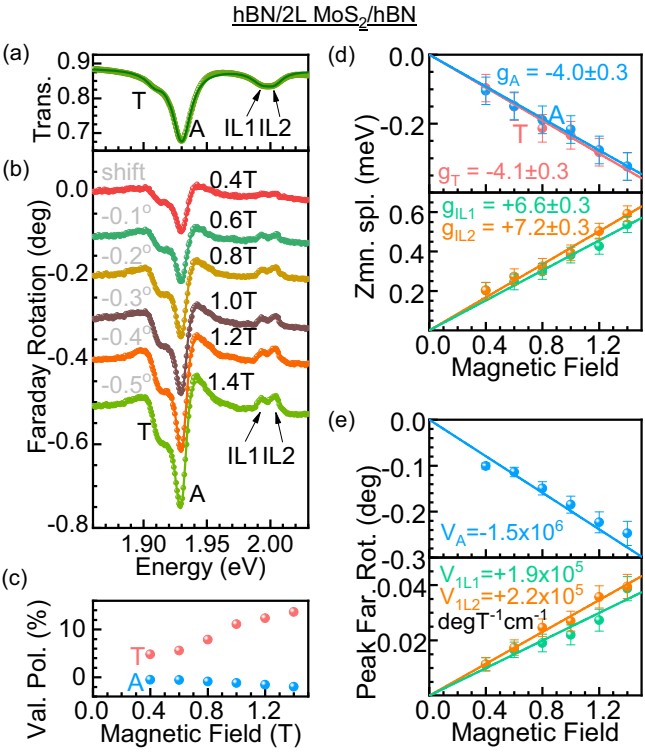

Fig. 3 | **Faraday rotation of intralayer and interlayer excitons in hBN-encapsulated bilayer MoS₂.** **a** Optical transmission spectrum of hBN/2 L MoS₂/hBN displaying trion T, intralayer exciton A, and a pair of interlayer excitons IL1 and IL2. **b** Experimental (spheres) and modeled (solid lines) Faraday rotation spectra of resonances in magnetic fields ranging from $B = 0.4\,T - 1.4\,T$. The spectra are vertically shifted by 0.1° successively for clarity. The shift is mentioned along with the respective plots. The characteristic line shapes of the T, A, and IL resonances are used for their assignment explained in the main text. **c** Magnetic-field-induced valley polarization of A and T derived from modeling in (**b**). **d** Valley Zeeman splittings deduced for the four resonances from the line shape modeling in (**b**). Linear fits (solid lines) are used to derive the effective $g$-factors $g_A$ and $g_T$. **e** Measured peak Faraday rotation of the resonances as a function of the magnetic field. Peak rotation is fitted linearly for deriving the Verdet constants $V_A = -(1.5 \pm 0.2) \times 10^6$ deg $T^{-1}$ cm$^{-1}$, $V_{IL1} = +(1.9 \pm 0.5) \times 10^5$ deg $T^{-1}$ cm$^{-1}$ and $V_{IL2} = +(2.2 \pm 0.5) \times 10^5$ deg $T^{-1}$ cm$^{-1}$.

to the high sensitivity of the technique (Fig. 3b). We believe that the reason for the appearance of the two IL features is the Stark effect splitting of the IL exciton due to a static electric field[67,68], which can be created by charge transfer from impurities in the substrate[69]. The Verdet constant of the interlayer excitons are $V_{IL1} = +(1.9 \pm 0.5) \times 10^5$ deg $T^{-1}$ cm$^{-1}$ and $V_{IL2} = +(2.2 \pm 0.5) \times 10^5$ deg $T^{-1}$ cm$^{-1}$ (Fig. 3e). In contrast, the intralayer exciton in this sample has a Verdet constant of $V_A = -(1.5 \pm 0.2) \times 10^6$ deg $T^{-1}$ cm$^{-1}$.

## Discussion

In Table 1, we compare the derived Verdet constants for our non-magnetic atomically thin semiconductors with conventional semiconductors, diluted magnetic semiconductors (DMSs), and magneto-optical elements used in optical isolators, covering the visible to infrared wavelength range[14,70]. Most importantly, we find that the Verdet constants for monolayer MoSe₂ and WSe₂ are larger by many orders of magnitude compared to other materials in this wavelength region[14,70]. A comparable magnitude is only known in the far-infrared (Terahertz) region, where graphene has a Verdet constant of $2.7 \times 10^7$ deg $T^{-1}$ cm$^{-1}$ (see ref. [52]), which is due to inter-Landau-level transitions (in the low-doping regime) or cyclotron resonances (high-doping regime) in the Terahertz region[52].

## Table 1 | Comparison of the Verdet constant of the excitons in our work with typical Verdet constants of excitons in semiconductors, diluted magnetic semiconductors (DMSs), and Faraday rotators used in optical isolators

| Verdet constant (units : deg T$^{-1}$ cm$^{-1}$) | | Wavelength (nm) |
| --- | --- | --- |
| **Peak Verdet constants of the A exciton in this work** | | |
| hBN/1L MoSe₂/hBN | $-(2.3 \pm 0.2) \times 10^7$ | 758 |
| hBN/1L WSe₂/hBN | $-(1.9 \pm 0.2) \times 10^7$ | 717 |
| hBN/2L MoS₂/hBN | $-(1.5 \pm 0.2) \times 10^6$ | 642 |
| **Peak Verdet constants of the interlayer (IL) exciton in this work** | | |
| hBN/2L MoS₂/hBN | $+(1.9 \pm 0.5) \times 10^5$ | 622 |
| | $+(2.2 \pm 0.5) \times 10^5$ | 619 |
| **Typical peak Verdet constants of exciton transitions in conventional semiconductors** | | |
| GaAs | 80[91], 130[92] | 1000[91], 860[92] |
| GaAs/AlGaAs multiple quantum wells | $4.6 \times 10^3$ (see ref. [93]) | 750[93] |
| GaSb | 20[1] | 620[1] |
| InSb | $0.7 \times 10^5$ (see ref. [1]) | 5300[1] |
| CdTe | $2.7 \times 10^3$ (see ref. [94]) | 775[94] |
| Si | 30[95] | 1050[95] |
| EuS | $4.6 \times 10^5$ (see ref. [96]) | 560[96] |
| **Typical Verdet constants of 2D magnetic materials** | | |
| Cr₂Te₂Ge₆ | $1.4 \times 10^5$ (see ref. [97]) | Not available |
| **Typical peak Verdet constants of diluted magnetic semiconductors** | | |
| Cd$_{0.6}$Mn$_{0.4}$Te | $1.0 \times 10^4$ (see ref. [98]) | 590[98] |
| GaMnAs | $7.0 \times 10^5$ (see ref. [99]) | 800[99] |
| CdMnHgTe | $1.0 \times 10^4$ (see ref. [100]) | 980[100] |
| **Typical Verdet constants of Faraday rotators used in optical isolators (see reviews, see refs. [14,70])** | | |
| Yttrium–Iron Garnet (YIG) | $0.9 \times 10^3$ (see ref. [101]) | 1107[101] |
| Bi-doped and Ce-doped YIG | $(4.4-5.0) \times 10^3$ (see ref. [102]) | 1550[102] |
| Terbium–Iron Garnet | $7.4 \times 10^3$ (see ref. [103]) | 1550[103] |
| Ce-doped Terbium–Iron Garnet (Ce:TbIG) | $2.9 \times 10^4$ (see ref. [104]), $0.6 \times 10^4$ (see ref. [103]) | 1550[104], 1550[103] |
| Bi-doped Terbium–Iron Garnet (Ce:TbIG) | $8.5 \times 10^3$ (see ref. [103]) | 1550[103] |

The materials cover the visible and infrared region.

In general, for the case of an ultrathin dielectric material with the thickness of our monolayer samples, Faraday rotation is expected to be negligibly small for energies far away from excitonic resonances. For example, the Faraday rotation of a 0.65 nm (typical thickness of a TMDC monolayer) thick InSb crystal in the sub-bandgap region at 0.23 eV under a magnetic field of $B = 1.4\,T$ (maximum applied field in this work) would be of the order of $10^{-7}$ deg[71]. However, close to an excitonic resonance, Faraday rotation is enhanced by many orders of magnitude. For instance, Faraday rotation for InAs at the exciton resonance is expected to be about $10^{-3}$ deg at the abovementioned conditions, which is four orders of magnitude larger[71]. In the case of TMDC monolayers, this value is enhanced even further by another 2–3 orders of magnitude leading to a giant Faraday effect. This is explained as follows. In general, Faraday rotation around exciton resonances occurs if both oscillator strength and Zeeman splitting of the exciton transition are non-zero. A large oscillator strength implies strong dips in transmission amplitudes $t_\pm$ for the $\sigma^\pm$ transitions (Eq. (4)). This results in an appreciable phase difference $\phi_+ - \phi_-$ and thereby a strong Faraday effect around the exciton transition. This enhancement of up

to four orders of magnitude in the Verdet constant around the exciton transitions, compared to the sub-bandgap region, has been confirmed earlier in III–V and II–VI semiconductors[1]. The observed giant excitonic Faraday rotation in TMDC monolayers is however due to a combined effect of (i) a giant exciton oscillator strength[21,72–75], and (ii) a large exciton g-factor ($\approx -4$[21,27,44,76]), when compared to conventional III–V and II–VI semiconductors[1,56,77]. The giant exciton oscillator strength in TMDCs is both due to the small excitonic spatial extent ($\approx 1$ nm) and the character of the electron and hole wavefunctions (localized in d-orbitals of the transition metals)[72–75,78]. Furthermore, considering the Wannier model, TMDCs are expected to have a large joint density of states at the van-Hove singularity at the K point due to a large exciton reduced mass[21,79,80], which is about 5–10 times larger compared to a typical III–V semiconductor such as GaAs[77,81]. This results in a giant exciton oscillator strength. The exciton g-factor of $\approx -4$ in TMDC monolayers is mainly due to the contributions of d-orbitals at the top of the valence bands at the K point, with modifications due to electron–hole interactions[21,27,76,78]. This is a large value compared to exciton g-factors in typical III–V 2D semiconductors i.e. quantum wells. For instance, the heavy-hole exciton g-factor in a GaAs/AlGaAs quantum well varies from −2 to +1 for well widths increasing from 2 to 25 nm[56,82]. The large exciton g-factor in a TMDC monolayer results in an appreciable lifting of the energetic degeneracy of the $\sigma^\pm$ polarized exciton transitions under a magnetic field. As a result, a strong Faraday effect is observed around the exciton resonance due to a large phase difference $\phi_\pm$ under a magnetic field (Eq. (5))[83]. We note that in the present work, Landau quantization effects[84–87] can be neglected due to the following reasons: (i) the large exciton binding energy (>150 meV) in hBN-encapsulated monolayers strongly dominates the magnetic quantization energy scale (i.e., $\hbar eB/2m^* \approx 0.3$ meV at $B = 1.4$ T) at the magnetic fields used[19,21,53]. This situation is unlike the case of graphene, where such Coulomb interactions are absent, and low magnetic fields (<5 T) are sufficient to observe Landau quantization in the THz regime[52]. (ii) Our hBN-encapsulated samples are not in the highly-doped regime, as evidenced by low (high) trion (exciton) oscillator strengths (Figs. 2 and 3). Inter-Landau transition effects in TMDCs have only been observed previously under high-doping conditions and large magnetic fields[85].

In conclusion, we have measured giant Faraday rotation around the A exciton transitions in hBN-encapsulated $MoSe_2$ and $WSe_2$ monolayers, as well as interlayer excitons in a $MoS_2$ bilayer. The Verdet constants are many orders of magnitude larger than those observed in conventional III–V or II–VI semiconductors and the well-known Faraday rotators used in optical isolators.

The Faraday rotation in monolayer TMDCs could be further enhanced by optimizing the multilayer structure of hBN and the TMDC monolayer by enhancing the dip in the transmittance in Figs. 2a and 3a. Furthermore, a heterostructure of a TMDC with a 2D ferromagnet could further raise the Verdet constant[20,88,89]. In such a heterojunction, strong magnetic exchange interaction effects between the ferromagnetic layer and the excitons in the TMDC are expected[90]. Finally, our work paves the way for a new generation of ultrathin optical polarization devices based on 2D materials.

## Methods

### Experimental setup for Faraday rotation spectroscopy on 2D materials

Broadband light from a Xe-arc lamp is collimated, and is linearly polarized, as shown in Fig. S1 of the Supplementary Information. It is focused on the sample mounted on the cold finger of a continuous-flow cryostat. The cold finger hangs between the pole pieces of an electromagnet, with a maximum magnetic field of $B = 1.4$ T applied perpendicular to the sample surface. The light undergoes a Faraday rotation $\phi_F$, and is reflected from a mirror. It is collimated after passing through a 10× long-working-distance infinity-corrected objective lens.

The collimated light is focused on a 20 μm diameter pinhole, which selects light from a spot of about 4 μm diameter on the sample surface. The light is again collimated and passes through a beam displacer, which spatially separates the linear polarization components. The two components are focused on the input slit of a 0.3 m focal length monochromator, are wavelength dispersed, and are collected simultaneously using a Peltier-cooled CCD detector. A Jones matrix analysis of the setup is performed to obtain the Faraday rotation as discussed in ref. 38.

### Determination of the complex dielectric tensor of hBN-encapsulated atomically thin semiconductors

The complex magneto-optical dielectric tensor is

$$\overleftrightarrow{\epsilon} = \begin{pmatrix} \epsilon_{xx1} + i\epsilon_{xx2} & \epsilon_{xy1} + i\epsilon_{xy2} \\ -(\epsilon_{xy1} + i\epsilon_{xy2}) & \epsilon_{xx1} + i\epsilon_{xx2} \end{pmatrix} \tag{7}$$

### Diagonal elements of the dielectric tensor

There are four unknowns in this matrix i.e. $\epsilon_{xx1}, \epsilon_{xx2}, \epsilon_{xy1}$ and $\epsilon_{xy2}$. Out of these, $\epsilon_{xx1}$ and $\epsilon_{xx2}$ can be determined by measuring reflectance $R(\widetilde{\epsilon}_{xx})$ and transmittance $T(\widetilde{\epsilon}_{xx})$ of the materials. The measured $R(\widetilde{\epsilon}_{xx})$ and $T(\widetilde{\epsilon}_{xx})$ spectra are shown in Fig. S2a, e for hBN-encapsulated $MoSe_2$ and $WSe_2$ monolayers in the Supplementary Information. Essentially, the experimentally measured $R(\widetilde{\epsilon}_{xx})$ and $T(\widetilde{\epsilon}_{xx})$ are simultaneously a function of the two unknowns $\epsilon_{xx1}$ and $\epsilon_{xx2}$[55]. A transfer-matrix-based approach is used to numerically calculate $\epsilon_{xx1}$ and $\epsilon_{xx2}$ with $R(\widetilde{\epsilon}_{xx})$ and $T(\widetilde{\epsilon}_{xx})$ spectra as inputs to the calculation[55]. The results of the calculation are presented in Fig. S2c, g of the Supplementary Information, respectively for the two materials.

### Off-diagonal elements of the dielectric tensor

The real and imaginary parts of the off-diagonal element $\widetilde{\epsilon}_{xy}$ can be determined from the measured Faraday rotation spectrum $\phi_F(E)$ as follows. $\phi_F(E)$ is the real part of the complex Faraday rotation spectrum $\widetilde{\phi}_F(E) = \phi_F + i\eta_F$ where $\eta_F$ is the Faraday ellipticity. Both $\phi_F$ and $\eta_F$ are functions of complex diagonal and off-diagonal components of the dielectric tensor, i.e., $\widetilde{\epsilon}_{xx}$ and $\widetilde{\epsilon}_{xy}$ (see Eq. (2)) which are four unknowns considering their real and imaginary parts. Since $\widetilde{\epsilon}_{xx}$ is already determined above (Fig. S2c, g of the Supplementary Information) for hBN-encapsulated $MoSe_2$ and $WSe_2$ monolayers, respectively, we are left with two unknowns, i.e., $\epsilon_{xy1}$ and $\epsilon_{xy2}$. The knowledge of $\phi_F$ and $\eta_F$ enable us now to calculate $\epsilon_{xy1}$ and $\epsilon_{xy2}$. Experimentally we measure only $\phi_F$. To determine $\eta_F$, we make use of the Kramers–Kronig analysis[61]. The real and imaginary parts of the complex Faraday effect given by $\widetilde{\phi}_F(E) = \phi_F + i\eta_F$ are related to each other through Kramers–Kronig dispersion relations for small values (i.e., $\tan \eta_F \sim \eta_F$) as follows:

$$\phi_F(\omega) = \frac{2}{\pi}\omega^2 P \int_0^\infty \frac{\text{arctanh} \eta_F(\omega')}{\omega'(\omega'^2 - \omega^2)} d\omega' \tag{8}$$

$$\eta_F(\omega) = \tanh\left[-\frac{2}{\pi}\omega P \int_0^\infty \frac{\phi_F(\omega')}{\omega'^2 - \omega^2} d\omega'\right] \tag{9}$$

Here P is the principal value, $\omega$ and $\omega'$ are the frequencies given as $E/\hbar$. An application of Eq. (9) on $\phi_F(E)$ spectra as input (solid lines in Fig. S2b, f of the Supplementary Information) yields Faraday ellipticity spectra $\eta_F(E)$ (dashed orange lines in Fig. S2b, f) for our samples. To test the applicability of our method, we use these dashed orange lines as input $\eta_F(E)$ spectra to calculate $\phi_F(E)$ using Eq. (8) (dashed blue lines) and compare with the experimentally measured $\phi_F(E)$ spectra (solid lines). The respective dashed blue and solid plots are in good agreement with each other, providing us confidence in our method. Finally, Eq. (2) is used to calculate the

real and imaginary parts of the off-diagonal dielectric tensor, $\epsilon_{xy1}$ and $\epsilon_{xy2}$. These are shown in Fig. S2d, h of the Supplementary Information and also in Figs. 2c and 3c of the main text for hBN-encapsulated $MoSe_2$ and $WSe_2$ respectively.

## Data availability
The minimum data sets that support the findings of this work have been deposited in the Figshare repository under the accession code https://figshare.com/s/06c6748220460582f8ff. Further requests for materials should be addressed to ashish.arora@iiserpune.ac.in.

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

## Acknowledgements

The authors acknowledge the financial support from the German Research Foundation (DFG project nos. AR 1128/1-1 and AR 1128/1-2), the Alexander von Humboldt Foundation, and NM-ICPS of the DST, Government of India through the I-HUB Quantum Technology Foundation (Pune, India), Project No. CRG/2022/007008 of SERB (Government of India), and MoE-STARS project No. MoE-STARS/STARS-2/2023-0912 (Government of India). Fruitful discussions with Thorsten Deilmann and Mukul Kabir are gratefully acknowledged.

## Author contributions

A.A. conceived the idea of the project. B.C. and A.A. performed the experiments and analyzed the data. N.K.W. and P.S. fabricated the samples. N.K.W., B.C., and A.A. built the setup. A.A. wrote the manuscript with contributions and suggestions from all authors. R.S., S.M.V., R.B., and A.A. supervised the project.

## Funding

## Competing interests

The authors declare no competing interests.
