## [Peer Review File · Nature Communications]

Giant Faraday rotation in atomically thin semiconductorsREVIEWER COMMENTS

Reviewer #1 (Remarks to the Author):

Since the experimental report on the Faraday rotation of monolayer graphene with the magnetic field, the magneto-optical responses of two-dimensional materials become the hot topic. In this work, the authors reported the experimental Faraday rotation of monolayer WSe_2 and MoSe_2 under the application of magnetic field. They attributed the giant Faraday rotation obtained here to the combined effect of a giant exciton oscillator strength and a large exciton g-factor. Their experimental works in this manuscript are valuable. However, I have some concerns about this work.

1. This work looks like the paper "Giant Faraday rotation in single- and multilayer graphene" *Nature Phys* 7, 48 (2010). In this paper, the giant Faraday rotation of monolayer graphene under the application of magnetic field is attributed to its inter-Landau-level transitions (in the low-doping regime) or cyclotron resonances (high-doping regime) in the terahertz region. In this work, the authors also used the magnetic field on monolayer WSe_2 and MoSe_2 to induce the Faraday rotation. From our understanding, the Landau levels still should exist in pristine monolayer WSe_2 and MoSe_2 when the magnetic field is applied [*Appl. Phys. Lett.* 105, 222411 (2014); *Nat. Commun.* 8, 1938 (); *Nat. Nano.* 12, 144 ()]. Why the Faraday rotation here is not from the inter-Landau level transition here but the combined effect of a giant exciton oscillator strength and a large exciton g-factor? Since the pristine monolayer WSe_2 and MoSe_2 have also possessed these kind of properties, but there is no magneto-optical response in the absence of magnetic field. The authors should clarify this point well.

2. The Faraday rotation basically comes from the nondiagonal term of the dielectric constant in the materials. We can see the information about the diagonal term of the dielectric constant in monolayer WSe_2 and MoSe_2 in the work but seldom information is found for the nondiagonal term of the dielectric constant in monolayer WSe_2 and MoSe_2 . That will help the reader understand it better.

3. The MO effects have been already discussed in monolayer TMDCs [see, e.g., *Phys. Rev. B* 100, 045411 (2019); *J. Appl. Phys.* 123, 034301 (2018); *Phys. Rev. B* 96, 125411 (2017)]. In particular, that would be more interesting if no magnetic field is required. There are some works working on this issue. Therefore, my impression is that the discussion about the MO effect in monolayer TMDC in the present manuscript is not quite new and it would be good for the authors to clarify their achievements in their work.

Reviewer #2 (Remarks to the Author):

In the paper “Giant Faraday rotation in atomically thin semiconductors” by Benjamin Carey et al, the authors study the Faraday rotation upon transmittance of visible light through monolayer transition metal dichalcogenides encapsulated in hBN. Using a recently introduced, home-built setup for broadband Faraday rotation spectroscopy, the degree of polarization rotation at moderate external magnetic fields can be investigated. Due to the large exciton oscillator strength and g-factor, record values for the Verdet constant are reported, which surpass the ones obtained for conventional semiconductors by two to three orders of magnitude. Using an exhaustive table summarizing previous literature values of Verdet constants, this key result is further highlighted. The data acquisition and subsequent analysis has carefully been carried out as also evidenced by the authors’ previous work introducing their experimental method. While the results are interesting and the record values for the Verdet constant are indeed impressive, I cannot support publication of the manuscript in Nature Communications in its current form. Please find my individual concerns listed below:

- The most important point concerns the degree of novelty of the work. In their previous paper (ref. 38, Carey, B. et al. High-Performance Broadband Faraday Rotation Spectroscopy of 2D Materials and Thin Magnetic Films. *Small Methods* 6, 2200885 (2022)), the authors have reported very similar experiments with almost identical plots. The only difference is the material system. From the data in Fig. 3e-g of this paper, a Faraday rotation of ~ 0.5 degree at the exciton resonance with an applied magnetic field of 1.2 T can be deduced. In this paper, the Verdet constant was simply not extracted and/or discussed, even though it should be very similar (up to a factor 3 or 4) to the values presented in the current manuscript.

- Could the authors comment on the reasons why WS₂ yields a slightly worse Verdet constant? Is the broader linewidth of the exciton resonance the main reason?

- Whereas the main insights such as the microscopic origin of the large Verdet constant (oscillator strength and g-factor) are nicely illustrated using MoSe₂ as an example, Figure 3 is hardly discussed. The underlying reason is the fact, that the second material system does

not contain any new information other than slightly different parameters of the exciton and hence a slightly different Verdet constant. In that sense, this comment is closely related to the first one concerning the novelty and impact. Most likely, the variation among different samples of the same type is even similar to the reported difference between MoSe₂ and WSe₂.

-I would suggest that the authors provide additional experimental data containing physics beyond the pure monolayer case. For example, this could encompass a 2D ferromagnet/TMDC heterostructure with yet higher Verdet constant as alluded to in the outlook. Another interesting scenario might be interlayer excitons, whose g-factor is yet higher than the monolayer excitons. The much smaller oscillator strength of the interlayer excitons compared to the monolayer might render this approach quite challenging, however.

-The authors argue that the Faraday effect has a plethora of applications. Consequently, a large Verdet constant is desirable. Yet the overall rotation cannot easily be scaled further based on the platform of monolayer TMDCs due to their atomically thin nature. Stacking multiple layers on top of each other would inevitably lead to interlayer hybridization and hence a reduction of the exciton binding energy. Thus, an hBN spacer layer is definitely required for incorporating more than one layer into a sample with an even higher Verdet constant. In short, if the experiments suggested in the previous comment prove to be too challenging to implement, an hBN/TMDC/hBN/TMDC/hBN structure might provide additional novel results beyond Figure 2.

-“Faraday rotation per using length”. Most likely the authors are referring to Faraday rotation per unit length.

Response to the reviewers' comments on manuscript NCOMMS-23-07643-T

We thank the referees for the detailed assessment of our manuscript. In the following, we address the concerns of the referees in a point-by-point response. All changes made to the manuscript are marked red in the revised text.

Referee 1.

Referee comment. *Since the experimental report on the Faraday rotation of monolayer graphene with the magnetic field, the magneto-optical responses of two dimensional materials become the hot topic. In this work, the authors reported the experimental Faraday rotation of monolayer WSe₂ and MoSe₂ under the application of magnetic field. They attributed the giant Faraday rotation obtained here to the combined effect of a giant exciton oscillator strength and a large exciton g-factor. Their experimental works in this manuscript are valuable.*

Our response. We thank the referee for finding our experimental work valuable.

Referee comment. *This work looks like the paper "Giant Faraday rotation in single- and multilayer graphene Nature Phys 7, 48 (2010). In this paper, the giant Faraday rotation of monolayer graphene under the application of magnetic field is attributed to its inter Landau-level transitions (in the low-doping regime) or cyclotron resonances (high-doping regime) in the terahertz region. In this work, the authors also used the magnetic field on monolayer WSe₂ and MoSe₂ to induce the Faraday rotation. From our understanding, the Landau levels still should exist in pristine monolayer WSe₂ and MoSe₂ when the magnetic field is applied [Appl. Phys. Lett. 105, 222411 (2014); Nat. Commun. 8, 1938 (); Nat. Nano.12, 144 ()]. Why the Faraday rotation here is not from the inter Landau level transition here but the combined effect of a giant exciton oscillator strength and a large exciton g-factor? Since the pristine monolayer WSe₂ and MoSe₂ have also possessed these kind of properties, but there is no magneto-optical response in the absence of magnetic field. The authors should clarify this point well.*

Our response. We thank the referee for this important comment. Our present work is fundamentally different from the work on graphene mentioned by the referee (*Nature Phys 7, 48 (2010)*). In the case of doped graphene, the 2D electron gas gets quantized to form Landau levels in the presence of a magnetic field. However, in our present work, Landau quantization of the excitonic transitions does not happen in undoped or slightly doped 2D semiconductors until under very high magnetic fields (>30 T). The reason for the relatively easy Landau quantization at low magnetic field in graphene is that the electron gas in graphene is free from excitonic effects. Therefore, the magnetic energy does not have to compete with any binding energy of electrons, such as the Coulomb energy in the case of excitons (see "*Physics of semiconductors in high magnetic fields*", N. Miura, Oxford University Press, 2008, for the phenomenon). As a result, quantization of the electron gas in graphene is possible under low values of the magnetic fields such as well below 7 T in *Nature Phys 7, 48 (2010)*. The energy of these inter-Landau transitions or cyclotron resonances falls in the Terahertz regime. Corresponding to these resonances, signatures are observed in the Faraday rotation spectrum in the Terahertz region of the spectrum.

In contrast, in hBN-encapsulated monolayers of WSe₂ and MoSe₂, we observe narrow excitonic transitions corresponding to the A excitons in the visible/infrared regime. The binding energy of these excitons is very large (>100 meV). For any Landau quantization effects to appear in the optical spectra, the magnetic energy has to compete with this binding energy. It turns out that one requires at least a few tens of Teslas of magnetic field, before any Landau quantization effects are experimentally observed in these systems (see *Arora et al., J. Appl. Phys. 129, 120902 (2021)* for a review on 2D materials).

The three papers mentioned by the referee and dealing with TMDCs *Appl. Phys. Lett.* 105, 222411 (2014), *Nat. Commun.* 8, 1938 (2017) and *Nat. Nano.* 12, 144 (2017) are not applicable to our work on the excitonic Faraday rotation due to the following reasons:

1) *Appl. Phys. Lett.* 105, 222411 (2014) theoretically discusses the creation of Landau Levels in TMDCs under very large magnetic fields (up to 50 T). Furthermore, it ignores the Coulomb interaction. Our work on excitonic Faraday rotation uses very low magnetic fields (<1.4T), and the Coulomb effects are extremely strong (>100 meV exciton binding energy). Therefore, we cannot compare our work to this paper.

2) *Nat. Commun.* 8, 1938 (2017) and *Nat. Nano.* 12, 144 (2017) are experimental works on highly-doped TMDC monolayers under magnetic fields. Under such conditions, the binding energy of the excitons and trions drastically reduces due to screening by the high-density electron gas. In fact, the features due to excitons and trions completely vanish under such conditions and inter-Landau level transitions are possible to be observed magneto-optically under large magnetic fields (from 4 T up to 9 T in *Nat. Nano.* 12, 144 (2017)). Our work does not involve highly-doped systems, and Faraday rotation measurements are performed under very low fields (<1.4 T). Therefore, we are working in a completely different regime and are therefore not able to directly compare our present work to these earlier works.

Indeed, the observed Faraday rotation around exciton lines is not due to Landau quantization effects. Instead, the intrinsic large Lande g factor of excitons (~ -4) and a giant exciton oscillator strength leads to a strong Faraday rotation around these transitions, which is explained as following. Due to the Zeeman splitting of the exciton line in a magnetic field, the material offers different dielectric functions to the left and right circularly polarized light around the exciton energy. This effect, when combined with a very narrow exciton transition due to a giant oscillator strength results in a large phase shift between the two circular polarization components of the incident linearly polarized light at the exciton energy. As a result, a giant Faraday rotation is observed due to these excitonic properties.

We have clarified this important point in our revised manuscript (marked in red), where we have also cited the works pointed out by the referee:

We note that in the present work, Landau quantization effects⁷⁸⁻⁸¹ can be neglected due to the following reasons: (i) the large exciton binding energy (> 150 meV) in hBN-encapsulated monolayers strongly dominates the magnetic quantization energy scale (i.e. $\hbar eB/2m^* \sim 0.3$ meV at $B = 1.4$ T) at the magnetic fields used^{19,21,82}. This situation is unlike the case of graphene, where such Coulomb interactions are absent, and low magnetic fields (<5 T) are sufficient to observe Landau quantization in the THz regime⁶⁶. (ii) Our hBN-encapsulated samples are not in the highly-doped regime, as evidenced by low (high) trion (exciton) oscillator strengths (Fig. 2 and 3). Inter-Landau transition effects have only been observed previously under high-doping conditions and large magnetic fields⁷⁹.

Referee comment. *The Faraday rotation basically comes from the nondiagonal term of the dielectric constant in the materials. We can see the information about the diagonal term of the dielectric constant in monolayer WSe2 and MoSe2 in the work but seldom information is found for the nondiagonal term of the dielectric constant in monolayer WSe2 and MoSe2. That will help the reader understand it better.*

Our response. We agree with the referee that Faraday rotation arises from the off-diagonal term of the complex dielectric tensor of the materials. For light incident along the z-direction perpendicular to the sample plane with $B \parallel z$ (i.e. Faraday geometry), the dielectric tensor can be written as (e.g. *Introduction of Modern Optics*, Grant R. Fowles, Dover Publications, New York, 1975, henceforth referred to as Fowles (1975)):

Fig. R1. **Calculation of the dielectric tensor.** See main text for details.

$$\vec{\epsilon} = \begin{pmatrix} \tilde{\epsilon}_{xx} & \tilde{\epsilon}_{xy} & 0 \\ -\tilde{\epsilon}_{xy} & \tilde{\epsilon}_{yy} & 0 \\ 0 & 0 & \tilde{\epsilon}_{zz} \end{pmatrix} \quad (\text{R1})$$

Here, each non-zero component is a complex quantity. In case of excitonic transitions in semiconductors, off-diagonal terms are normally zero in the absence of magnetic field, and are activated in the presence of a magnetic field due to the exciton Zeeman splitting. The complex off-diagonal component $\tilde{\epsilon}_{xy}$ is given as (Fowles (1975))

$$\tilde{\epsilon}_{xy}(B) = \epsilon_{xy1} + i\epsilon_{xy2} = \frac{\tilde{V}\lambda_0\tilde{n}_{xx}B}{\pi} \quad (\text{R2})$$

where $\tilde{V} = V_{FR} + iV_{FE}$ is the complex Verdet constant, having its components V_{FR} and V_{FE} related to Faraday rotation and Faraday ellipticity, $\tilde{n}_{xx} = n + ik = \sqrt{\tilde{\epsilon}_{xx}} = \sqrt{\epsilon_{xx1} + i\epsilon_{xx2}}$ is the complex refractive index. The calculation of $\tilde{\epsilon}_{xy}(B)$ is a non-trivial task. Eq. R2 suggests that for calculating $\tilde{\epsilon}_{xy}$ at a given magnetic field B , we require the knowledge of the complex diagonal refractive index \tilde{n}_{xx} and complex Verdet constant \tilde{V} .

For \tilde{n}_{xx} , we measure reflectance $R(E)$ and transmittance $T(E)$ spectra of our hBN-encapsulated monolayer MoSe₂ and WSe₂ samples at $T = 10$ K. We plot $R(E)$ and $1 - T(E)$ spectra in Fig. R1 a and e for the two samples, respectively. Since $R(E)$ and $T(E)$ are a function of (n, k) or in turn $(\epsilon_{xx1}, \epsilon_{xx2})$, we derive these quantities numerically using a transfer-matrix method-based analysis (see Fig. R1 c and g). Therefore, \tilde{n}_{xx} in eq. R2 is derived.

For $\tilde{V}(E) = V_{FR}(E) + iV_{FE}(E)$, $V_{FR}(E) = \frac{\phi_F(E)}{dB}$ is experimentally measured (Eq. 6 of the main text). For $V_{FE}(E)$, we make use of the fact that V_{FR} and V_{FE} are related to each other through Kramers-Kronig relationships as follows (P. Kielar, J. Opt. Soc. Am. B **11**, 854 (1994)):

$$V_{FR}(\omega) = \frac{2}{\pi} \omega^2 \mathbf{P} \int_0^\infty \frac{\arctanh V_{FE}(\omega')}{\omega'(\omega'^2 - \omega^2)} d\omega', \quad (\text{R3})$$

$$V_{FE}(\omega) = \tanh \left[-\frac{2}{\pi} \omega \mathbf{P} \int_0^\infty \frac{V_{FR}(\omega')}{\omega'^2 - \omega^2} d\omega' \right] \quad (\text{R4})$$

where \mathbf{P} represents the principal value of the integral, and $E = \hbar\omega$. Using eq. R4, $V_{FE}(E)$ is determined. Faraday ellipticity for $B = 1.4\text{T}$ is plotted in Fig. R1 b and f (dashed orange line). To test if the derived Faraday ellipticity is correct, we used eq. (R3) to derive the Faraday rotation using the dashed orange line as the input. The calculated spectrum (dashed blue line) is in agreement with the experimental data within 10% error, giving us confidence about our method. Hence, $\tilde{V}(E)$ in eq. (R2) is obtained.

Finally, eq. (R2) is used to calculate the complex off-diagonal dielectric constant $\tilde{\epsilon}_{xy}$. Real and imaginary parts of the $\tilde{\epsilon}_{xy}$ i.e. ϵ_{xy1} and ϵ_{xy2} are plotted in Fig. R1 d and h for $B = 1.4\text{T}$.

In the main text, we have introduced the complex dielectric tensor as well as complex Verdet constant on page 2. We have modified Fig. 2 and Fig. 3 to include the off-diagonal dielectric constant as Figs. 2c and 3c. We include the following text in the main manuscript file on page 3: “From our data, we determine the real and imaginary components (ϵ_{xy1} and ϵ_{xy2}) of the off-diagonal dielectric constant $\tilde{\epsilon}_{xy}$ of monolayer WSe₂ as well following eq. 2. An example of ϵ_{xy1} and ϵ_{xy2} for $B = 1.4\text{T}$ is shown in Fig. 2c. As required by eq. 2, the procedure involves a calculation of Faraday ellipticity using a Kramers-Kronig analysis of our data in Fig. 2b⁵⁹, as well as the complex diagonal dielectric function $\tilde{\epsilon}_{xx}$. The details of the calculation are provided in the supporting information.”

The detailed process to extract $\tilde{\epsilon}_{xy}$ has been included in the supporting information of the revised manuscript. We thank the referee for this comment since it has improved its quality and also underlines the novelty of our manuscript.

Referee comment. The MO effects have been already discussed in monolayer TMDCs [see, e.g., Phys. Rev. B 100, 045411 (2019); J. Appl. Phys. 123, 034301 (2018); Phys. Rev. B 96, 125411 (2017)]. In particular, that would be more interesting if no magnetic field is required. There are some works working on this issue. Therefore, my impression is that the discussion about the MO effect in monolayer TMDC in the present manuscript is not quite new and it would be good for the authors to clarify their achievements in their work.

Our response. We thank the referee for suggesting these important references. We notice that all of the references pointed out by the referee describe magneto-optical effects in 2D semiconductors from a theoretical point of view. *Phys. Rev. B 100, 045411 (2019)* discusses the theoretically expected Verdet constants of excitons in monolayer MoS₂ and WSe₂ with values reaching up to 5×10^6 rad/T per monolayer thickness, i.e. 4.4×10^3 deg T⁻¹cm⁻¹. Our work experimentally measures the Verdet constant of excitons in monolayer TMDCs for the first time. Furthermore, we find that the values which we actually measure are about 4 orders of magnitude higher than the theoretically predicted values in *Phys. Rev. B 100, 045411 (2019)*. We thank the referee for pointing out these papers which we have now cited in the revised version of the manuscript.

We notice that in the absence of a magnetic field in a bare semiconductor, the valley Zeeman splitting and valley polarization are zero. However, one can create a heterostructure of a 2D semiconductor such as WSe₂ with a 2D magnet such as CrI₃ and induce a magnetic-exchange interaction from the 2D magnet to the 2D semiconductor (e.g. Zhong et al., *Sci. Adv.* 3, e1603113 (2017).). In this case, magneto-optical effects such as Faraday/Kerr effects are expected even in the absence of an applied external magnetic field. However, experimentally realizing such a heterostructure system of high quality is extremely challenging and part of our future works. We have included this in the last paragraph of the main text of our manuscript: “Furthermore, a heterostructure of a TMDC with a 2D ferromagnet could further raise the

Verdet constant^{20,86,87}. In such a heterojunction, strong magnetic exchange interaction effects between the ferromagnetic layer and the excitons in the TMDC are expected⁸⁸.”

Referee 2.

Referee comment. *In the paper “Giant Faraday rotation in atomically thin semiconductors” by Benjamin Carey et al, the authors study the Faraday rotation upon transmittance of visible light through monolayer transition metal dichalcogenides encapsulated in hBN. Using a recently introduced, home-built setup for broadband Faraday rotation spectroscopy, the degree of polarization rotation at moderate external magnetic fields can be investigated. Due to the large exciton oscillator strength and g-factor, record values for the Verdet constant are reported, which surpass the ones obtained for conventional semiconductors by two to three orders of magnitude. Using an exhaustive table summarizing previous literature values of Verdet constants, this key result is further highlighted. The data acquisition and subsequent analysis has carefully been carried out as also evidenced by the authors’ previous work introducing their experimental method.*

Our response. We thank the referee for the detailed report on our manuscript, and appreciating the careful data acquisition and analysis in our work.

Referee comment. *-The most important point concerns the degree of novelty of the work. In their previous paper (ref. 38, Carey, B. et al. High-Performance Broadband Faraday Rotation Spectroscopy of 2D Materials and Thin Magnetic Films. Small Methods 6, 2200885 (2022)), the authors have reported very similar experiments with almost identical plots. The only difference is the material system. From the data in Fig. 3e-g of this paper, a Faraday rotations of ~0.5 degree at the exciton resonance with an applied magnetic field of 1.2 T can be deduced. In this paper, the Verdet constant was simply not extracted and/or discussed, even though it should be very similar (up to a factor 3 or 4) to the values presented in the current manuscript.*

Our response. We thank the referee for the comment. Our earlier work (Carey et al., Small Methods 2022) involved a detailed description of a new Faraday rotation spectroscopy (FRS) technique, which we developed for performing spatially-resolved temperature-dependent FRS measurements. We demonstrated the functionality of the method by performing exemplary FRS measurements on an hBN-encapsulated WS₂ monolayer and a TbFe₀ ferrimagnetic alloy. The referee is correct that we did not focus on the Verdet constants of the A exciton i.e. $-(4.6 \pm 0.2) \times 10^6$ of the WS₂ layer in this earlier manuscript. As discussed in the next section, this value is found to be a factor of 4 to 6 smaller than the WSe₂ and MoSe₂ monolayer samples in the present work due to the relatively poor sample quality of the hBN-encapsulated WS₂ monolayer used in Carey et al., Small Methods, 2022.

We believe that the topic of giant Verdet constants in the visible and infrared region of the spectrum in high-quality monolayers deserves a focused attention due to the fundamental scientific importance and its potential in opening a new route to ultrathin magneto-optical device-based applications. Only a separate manuscript with high visibility would justify this. It also requires the best sample qualities. Therefore, for the present work, we tried hard and succeeded in producing high-quality hBN-encapsulated WSe₂ and MoSe₂ monolayers on sapphire substrates, with A exciton line widths approaching the homogeneous limit.

We further address the comments of the referee concerning the novelty of our work as follows:

- 1) We perform additional optical spectroscopy measurements such as reflectance and transmittance on our samples simultaneously. Using these in conjunction with our measured Faraday rotation spectra, we calculate the complete complex dielectric tensor of the two representative materials monolayer WSe₂ and MoSe₂ from the van der Waals semiconductors family for the first time. We include the real and imaginary parts of the diagonal and off-diagonal dielectric elements of the tensor in the revised version (see above for details, answer

to referee 1). Knowledge of the dielectric tensor is important for designing and predicting the magneto-optical response of heterostructures of 2D semiconductors.

2) We perform additional Faraday rotation spectroscopy measurements on interlayer excitons in a hBN-encapsulated MoS₂ bilayer for the first time, which have a positive and higher absolute g-factor than intralayer excitons, and include them in our revised manuscript (see further below).

Following these further improvements in the manuscript, we are confident that our work deserves publication in a high-quality journal such as Nature Communications.

Referee comment. - *Could the authors comment on the reasons why WS₂ yields a slightly worse Verdet constant? Is the broader linewidth of the exciton resonance the main reason?*

- *Whereas the main insights such as the microscopic origin of the large Verdet constant (oscillator strength and g-factor) are nicely illustrated using MoSe₂ as an example, Figure 3 is hardly discussed. The underlying reason is the fact, that the second material system does not contain any new information other than slightly different parameters of the exciton and hence a slightly different Verdet constant. In that sense, this comment is closely related to the first one concerning the novelty and impact. Most likely, the variation among different samples of the same type is even similar to the reported difference between MoSe₂ and WSe₂.*

Our response. These two points are related, therefore we are addressing them together. The referee is correct. All the three materials WS₂, WSe₂ and MoSe₂ are qualitatively similar concerning their exciton Verdet constants. In Fig. R2 a and b, we compare the optical transmittance spectra of the three hBN-encapsulated monolayer samples. An overall difference in the transmission between the three materials is due to the different extent of interference effects due to encapsulation with hBN of dissimilar thicknesses. There are quantitative differences in the A exciton oscillator strengths of the three materials (Table 1). We find that the A exciton oscillator strength is the largest in WS₂, followed by WSe₂ and MoSe₂ respectively. However, WS₂ yields a slightly worse Verdet constant compared to WSe₂ and MoSe₂ in our samples. The derived A exciton full-width at half-maximum (FWHM) line widths are compared in Table 1. We find that the FWHM line width for the WS₂ case is about a factor of 2 larger than WSe₂ and MoSe₂, even though the oscillator strength parameters are comparable. Therefore, the poor quality of the WS₂ layer results in a lower Verdet constant.

Fig. R2. **Comparison of the transmittance spectra of the three materials.** (a) Comparison of transmittance of hBN-encapsulated WSe₂ and (b) hBN-encapsulated MoSe₂ with hBN-encapsulated WS₂ (from Carey et al., Small Methods 2022) on the sapphire substrates. We notice that WSe₂ and MoSe₂ samples have a deeper transmission dip compared to WS₂. The reason is a larger exciton line width in our WS₂ sample (see Table 1 of this document).

We also notice that although the A exciton oscillator strength in the MoSe₂ sample is the smallest among the three samples, its Verdet constant is the largest due to the narrowest line width in this sample. Additionally, we agree with the referee that there will be a variation found in the samples of the same type depending on A-exciton line widths.

Table 1 Comparison of the FWHM line widths and the oscillator strength parameters of the three samples

2D material	FWHM (meV)	Osc. Str. Parameter (arb. units)	A exciton Verdet constant (deg T ⁻¹ cm ⁻¹)
hBN-encapsulated WS ₂	6.3	3.5	$-(4.6 \pm 0.2) \times 10^6$
hBN-encapsulated WSe ₂	3.4	3.1	$-(1.9 \pm 0.2) \times 10^7$
hBN-encapsulated MoSe ₂	2.7	2.5	$-(2.3 \pm 0.2) \times 10^7$

Referee comments. I would suggest that the authors provide additional experimental data containing physics beyond the pure monolayer case. For example, this could encompass a 2D ferromagnet/TMDC heterostructure with yet higher Verdet constant as alluded to in the outlook. Another interesting scenario might be interlayer excitons, whose g-factor is yet higher than the monolayer excitons. The much smaller oscillator strength of the interlayer excitons compared to the monolayer might render this approach quite challenging, however.

The authors argue that the Faraday effect has a plethora of applications. Consequently, a large Verdet constant is desirable. Yet the overall rotation cannot easily be scaled further based on the platform of monolayer TMDCs due to their atomically thin nature. Stacking multiple layers on top of each other would inevitably lead to interlayer hybridization and hence a reduction of the exciton binding energy. Thus, an hBN spacer layer is definitely required for incorporating more than one layer into a sample with an even higher Verdet constant. In short, if the experiments suggested in the previous comment prove to be too challenging to implement, an hBN/TMDC/hBN/TMDC/hBN structure might provide additional novel results beyond Figure 2.

Our response. We thank the referee for the suggestions and giving us further ideas to incorporate in the manuscript concerning novelty. The referee provides us three interesting experiments, which are the following:

- 1) FR spectroscopy of 2D ferromagnet/TMDC heterostructures
- 2) FR spectroscopy of interlayer excitons whose g-factor is larger, but the oscillator strength is smaller than the intralayer excitons
- 3) FR spectroscopy of an hBN/TMDC/hBN/TMDC/hBN heterostructure

Out of these ideas, while trying 1) is extremely challenging, we have tried 2) and 3). Here we summarize our results:

Idea 2). FR spectroscopy of interlayer excitons in a MoS₂ bilayer. We performed Faraday rotation spectroscopy on hBN-encapsulated bilayer MoS₂ and included our results in the revised main manuscript as Fig. 3, reproduced as Fig. R3 (a – d) in the rebuttal. It was a challenging measurement and analysis due to the following reasons: (i) Preparation of the high-quality samples with narrow interlayer exciton lines is challenging. (ii) Our bilayer MoS₂ samples are unintentionally doped. This results in the appearance of a charged exciton line T, about 21 meV below the neutral exciton A (Fig. R3 a and b). This makes the analysis of the FR spectroscopy data difficult, since apart from the valley Zeeman splitting, charged excitons also polarize strongly in the presence of a magnetic field (Koperski et al., *2D Materials*, 6,

Figure R3. Faraday rotation of intralayer and interlayer excitons in hBN-encapsulated bilayer MoS₂. a) Optical transmission spectrum of hBN/2L MoS₂/hBN displaying a trion T, an intralayer exciton A, and a pair of interlayer excitons IL1 and IL2. b) Experimental (spheres) and modelled (solid lines) Faraday rotation spectra of the resonances in magnetic fields ranging from $B = 0.4 \text{ T} - 1.4 \text{ T}$. The spectra are vertically shifted successively by 0.1° for clarity. The shift is mentioned along with the respective plots. The characteristic line shapes of the T, A and IL resonances are used for their assignment explained in the main text. c) Valley Zeeman splittings deduced for the four resonances from the line shape modelling in (b). Linear fits (solid lines) are used to derive the effective g-factors g_A and g_T . d) Measured peak Faraday rotation of the resonances as a function of the magnetic field. The peak rotation is fitted linearly for deriving the Verdet constants $V_A = -(1.5 \pm 0.2) \times 10^6 \text{ deg T}^{-1} \text{ cm}^{-1}$, $V_{IL1} = +(1.9 \pm 0.5) \times 10^5 \text{ deg T}^{-1} \text{ cm}^{-1}$ and $V_{IL2} = +(2.2 \pm 0.5) \times 10^5 \text{ deg T}^{-1} \text{ cm}^{-1}$. (e) Magnetic-field-induced valley polarization of the trion-exciton pair in the MoS₂ bilayer. The inverted valley polarization of the trion-exciton pair supports their observation in optical transmittance and Faraday rotation spectra.

015001 (2018)). In addition, we notice two close-lying interlayer excitons IL1 and IL2, separated by $\sim 10 \text{ meV}$. Therefore, the line shape modelling involves a transfer-matrix-based simulation of 4 resonances in both optical transmittance and Faraday rotation with Zeeman splittings and valley polarizations as the free parameters. We add the following paragraph to the revised main manuscript and the supporting information to explain our findings:

In the main manuscript file p. 3: **As an example of a material with a positive Verdet constant, we perform FR spectroscopy of interlayer excitons (IL) in a hBN-encapsulated MoS₂ bilayer. Interlayer excitons in bilayer and bulk TMDCs are known to have a positive g-factor which is opposite in sign to the intralayer excitons^{21,60,61}. However, interlayer excitons have a much smaller oscillator strength and a larger line width compared to the intralayer excitons⁶⁰⁻⁶². Therefore, their Verdet constant is expected to be smaller. The measured transmittance (for $B = 0$) and the Faraday rotation spectra (under $B = 0.4 - 1.4 \text{ T}$) of the hBN-encapsulated MoS₂ bilayer sample are shown in Fig. 3(a). Clear signatures corresponding to the intralayer exciton (A) at 1.930 eV , intralayer trion (T) at 1.909 eV , and a split interlayer exciton (IL₁ at 1.994 eV and IL₂ at 2.004 eV) are visible in the Faraday rotation spectra. The assignment of**

the features are performed on the following grounds: A and T resonances have similar g -factors, nearly equal to -4 suggesting their intralayer character^{21,61}. T polarizes strongly under magnetic field, with its valley polarization approaching $\sim +14\%$ (Fig. S3 of the supporting information). Furthermore, A polarizes only weakly (valley polarization $\sim -2\%$). Large polarization of T with an opposite sign compared to A is characteristic for the appearance of a trion-exciton pair⁶³. We notice that the binding energy of the trion in the MoS₂ bilayer is about 21 meV. In comparison, the reported value in a non-encapsulated bilayer is 27 meV⁶⁴. A smaller value in our work signifies the effect of an increased dielectric constant around the trion, due to hBN encapsulation. The split IL exciton lines are identified due to their positive g -factors ($g_{\text{IL1}} = +6.6 \pm 0.3$ and $g_{\text{IL2}} = +7.2 \pm 0.3$)^{21,60,61}. In previous works, one IL resonance has been observed in optical reflectance spectra^{21,61}. In our transmittance spectra, we also notice one (broad) IL line (Fig. 3a), while Faraday rotation spectroscopy is able to resolve two close-lying IL features due to the high sensitivity of the technique (Fig. 3b). We believe that the reason for the appearance of the two IL features is the Stark effect splitting of the IL exciton due to a static electric field^{65,66}, which can be created by charge transfer from impurities in the substrate⁶⁷. The Verdet constant of the interlayer excitons are $V_{\text{IL1}} = +(1.9 \pm 0.5) \times 10^5 \text{ deg T}^{-1} \text{ cm}^{-1}$ and $V_{\text{IL2}} = +(2.2 \pm 0.5) \times 10^5 \text{ deg T}^{-1} \text{ cm}^{-1}$. In contrast, the intralayer exciton in this sample has a Verdet constant of $V_{\text{A}} = -(1.5 \pm 0.2) \times 10^6 \text{ deg T}^{-1} \text{ cm}^{-1}$.

In the revised supporting information, p.3.: **“Assignment of the trion-exciton pair in the hBN-encapsulated MoS₂ bilayer.** The Faraday rotation spectra are modelled using a transfer-matrix-based method for $B = 0.4 \text{ T} - 1.4 \text{ T}$ [1]. The modelled spectra are shown as solid lines in Fig. 3(b) of the main text. The magnetic-field-induced valley polarization of the neutral and charged exciton (A and T, respectively) is plotted in Fig. S3. The opposite degree of the polarization with magnetic field is a well-established evidence for the occurrence of a trion-exciton pair in the literature of semiconductor quantum wells [5]. Therefore, we assigned the first two resonances in our transmittance/FR spectra to a neutral and a charged exciton. Our assignment also agrees with an earlier report [6].”

Idea 3). FR spectroscopy of a hBN/TMDC/hBN/TMDC/hBN heterostructure. For this experiment, we fabricated a few hBN/WSe₂/hBN/WSe₂/hBN samples. The goal was to obtain a sample where the A excitons have homogeneous line widths, and are at the same time energetically degenerate in the two individual WSe₂ monolayers. However, this experiment has proven extremely challenging. It is because, normally, in creating hBN-encapsulated heterostructures using the standard mechanical exfoliation process, one finds only extremely small regions ($\sim 1 - 2 \mu\text{m}$ across) with homogeneous line widths. The remaining interfacial areas are either mostly covered with bubble-like regions, which collect dirt between the interfaces, or have a broad exciton line width due to imperfect encapsulation. In an hBN/TMDC/hBN/TMDC/hBN heterostructure, there are four interfacial junctions. The probability that there will be a clean interface with an identical quality for all of the four junctions of the heterostructure, resulting in two overlapping homogeneously broadened excitons is extremely small. In our trials of a few hBN/WSe₂/hBN/WSe₂/hBN samples which we created and tested, we find identical results as follows: The exciton lines are not very narrow. The best exciton line widths are $\sim 10 \text{ meV}$ compared to a homogeneous line width of $\sim 3 - 4 \text{ meV}$ in a single hBN-encapsulated monolayer. Furthermore, although the optical transmission shows two spectrally overlapping A exciton lines, the FR spectra always show two close-lying but separate exciton lines, corresponding to each of the two WSe₂ monolayers. We have not found a location on the samples where the two excitons exactly spectrally overlap. Due to these reasons, the Faraday rotation is not found to be as large as our hBN-encapsulated monolayers. Figure R4 shows one set of results we obtained. The deduced Verdet constant is $V_{\text{A1}} = -1.9 \times 10^6 \text{ deg T}^{-1} \text{ cm}^{-1}$.

Figure R4. Faraday rotation spectroscopy of a hBN/WSe₂/hBN/WSe₂/hBN heterostructure. a) Experimental (green spheres) and modelled (green solid line) optical transmission spectrum of an hBN/WSe₂/hBN/WSe₂/hBN heterostructure displaying two exciton lines A1 and A2. Dashed lines are the fits for the individual excitons. b) Faraday rotation spectra for $B = 0.4 \text{ T}$, 0.7 T and 0.9 T . The spectra are vertically shifted by 0.1° successively for clarity. (c) Peak Faraday rotation for the A1 exciton line. The Verdet constant obtained from the linear fit is $V_{A1} = -1.9 \times 10^6 \text{ degT}^{-1}\text{cm}^{-1}$.

Referee comment: "Faraday rotation per using length". Most likely the authors are referring to Faraday rotation per uNIT length.

Our response. We thank the referee for pointing out this typo error. We have corrected this in the revised version.

REVIEWER COMMENTS

Reviewer #1 (Remarks to the Author):

1. The authors gave the explanations on "their Faraday rotation here is not from the inter Landau level transition here but the combined effect of a giant exciton oscillator strength and a large exciton g-factor." are "Due to the Zeeman splitting of the exciton line in a magnetic field, the material offers different dielectric functions to the left and right circularly polarized light around the exciton energy. This effect, when combined with a very narrow exciton transition due to a giant oscillator strength results in a large phase shift between the two circular polarization components of the incident linearly polarized light at the exciton energy." They are more like the classical explanations to the Faraday rotation other than the microscopic mechanism. To prove this, the authors should provide more convincing facts.

2. Authors claim: " However, in our present work, Landau quantization of the excitonic transitions does not happen in undoped or slightly doped 2D semiconductors until under very high magnetic fields (>30 T)..". There are many theoretical works on the LL of monolayer TMDC under the application of magnetic field. Regarding to the magnitude of magnetic field, there is also the theoretical works on the LL in monolayer MoS₂ quantum Hall systems [PHYSICAL REVIEW B 90, 045427 (2014)], where magnetic field is around 5T. Why the authors can achieve the good FR at small magnetic field (1.4T)? And if small magnetic field applied here is their innovation point, the authors should present these in the introduction part.

Reviewer #2 (Remarks to the Author):

In the revised manuscript, the authors have satisfactorily addressed all my comments. In particular this concerns the degree of novelty: The authors have clarified that the improved sample quality represents a major step forward compared to their previous work. I agree that this warrants a separate high-impact publication focusing not on the experimental method but rather on the intrinsic material properties. The new data sets that have been included in the revised manuscript also add to this point. Combined with the new discussion about the off-diagonal elements of the dielectric tensor, the Verdet constant of opposite

sign for the interlayer excitons definitely provide a more complete picture about the Faraday rotation in transition metal dichalcogenides now. I appreciate the fact that the authors have tried to implement heterostructures with a hBN spacer to reach yet higher Verdet constants and fully understand that this task has been limited by multiple imperfect interfaces. Overall, I can now fully recommend the manuscript for publication in Nature Communications.

Response to the reviewers' comments on manuscript NCOMMS-23-07643-T

We thank both referees for reading our revised manuscript and our clarifications. We are happy that reviewer 2 recommends publication in Nature Communications. Here, we address the remaining concerns of reviewer 1 in a point-by-point fashion:

Referee 1.

Referee comment. *The authors gave the explanations on "their Faraday rotation here is not from the inter Landau level transition here but the combined effect of a giant exciton oscillator strength and a large exciton g-factor." are "Due to the Zeeman splitting of the exciton line in a magnetic field, the material offers different dielectric functions to the left and right circularly polarized light around the exciton energy. This effect, when combined with a very narrow exciton transition due to a giant oscillator strength results in a large phase shift between the two circular polarization components of the incident linearly polarized light at the exciton energy." They are more like the classical explanations to the Faraday rotation other than the microscopic mechanism. To prove this, the authors should provide more convincing facts.*

Our response. As mentioned in our original submission, we indeed believe that the excitonic Faraday rotation in semiconducting transition metal dichalcogenides (TMDCs) such as MoS₂, MoSe₂, WS₂ and WSe₂ is due to a combined effect of a giant exciton oscillator strength and a large exciton g factor. In support of this explanation from a microscopic perspective, we had originally included the following text on page 5 of the manuscript along with the supporting references:

"The giant exciton oscillator strength in TMDCs is both due to the small excitonic spatial extent (~1 nm) and the character of the electron and hole wavefunctions (localized in d-orbitals of the transition metals) [71–74,77].

Furthermore, considering the Wannier model, TMDCs are expected to have a large joint density of states at the van-Hove singularity at the K point due to a large exciton reduced mass[21,78,79], which is about 5-10 times larger compared to a typical III-V semiconductor such as GaAs[76,80]. This results in a giant exciton oscillator strength. The exciton g-factor of approximately -4 in TMDC monolayers is mainly due to the contributions of d-orbitals at the top of the valence bands at the K point, with modifications due to electron-hole interactions [21,27,75,77]. This is a large value compared to exciton g-factors in typical III-V 2D semiconductors i.e. quantum wells. For instance, the heavy-hole exciton g-factor in a GaAs/AlGaAs quantum well varies from -2 to +1 for well widths increasing from 2 nm to 25 nm[54,81]. The large exciton g-factor in a TMDC monolayer results in an appreciable lifting of the energetic degeneracy of the σ^\pm polarized exciton transitions under a magnetic field. As a

Figure R1. *Left panel:* Spin-orbit-split conduction and valence bands (CB and VB) at the K^\pm points of the Brillouin zone of a TMDC monolayer, such as MoSe₂, in the absence of a magnetic field. Δ_c and Δ_v denote the spin-orbit splitting in CB and VB. *Right panel:* Coupling of spin and atomic orbital magnetic moments at the top of the VB and the bottom of the CB under an external magnetic field resulting in an energy shift of bands depicted by blue arrows. This lifts the energetic degeneracy of the σ^\pm transitions at the K^\pm points. The valley Zeeman splitting is given as $\Delta E = 2(\Delta E_{cb} - \Delta E_{vb}) \sim g_X \mu_b B$, where g_X is the exciton g factor.

result, a strong Faraday effect is observed around the exciton resonance due to a large phase difference ϕ_{\pm} under a magnetic field (Eq. 5).”

To further address the referee’s concern, we now present our arguments in a more detailed fashion following the literature^{1–6}, by using a MoSe₂ monolayer as an example. The optical transitions relevant to the present work take place at the K^{\pm} points (‘valleys’) of the hexagonal Brillouin zone of the TMDC monolayer^{3,4}. Due to an absence of inversion symmetry, and a strong spin-orbit interaction in a TMDC monolayer, the top (bottom) of the valence (conduction) bands at the K^+ and K^- points are associated with opposite signs of magnetic moments^{1,5–8}(Fig. R1). Due to this, the optical transitions at the K^+ and K^- points possess opposite helicity of circular polarization i.e. σ^+ and σ^- respectively^{6,9–11}. In the absence of a magnetic field, the two transitions are energetically degenerate⁵ (Fig. R1). However, in the presence of a magnetic field, the energies of the conduction and valence bands involved in the exciton transition are modified via magnetic coupling the magnetic moments. Since the two valleys have opposite magnetic moments, they couple with the magnetic field in an opposite manner, moving bands at the K^+ and K^- points in the opposite directions (Fig. R1). These shifts result in breaking the degeneracy of the σ^{\pm} optical transitions, i.e. a so-called ‘valley Zeeman splitting’ of the exciton, $\Delta E = E_{\sigma^+} - E_{\sigma^-} = 2(\Delta E_{cb} - \Delta E_{vb}) \sim g_X \mu_b B$, where g_X is the exciton g factor^{5–7,12}. Experimentally, the g factors of the excitons and trions in monolayer TMDCs are observed to be approximately equal to -4 ^{5,6,12}. Direct microscopic calculations of the exciton g factor on the basis of *GW*-BSE *ab initio* theory predicts the values of the exciton g factors very close to -4 ¹. Other groups have also calculated the interband-transition g factors of similar magnitudes without including excitonic effects^{8,13,14}.

Faraday rotation of the linearly polarized light passing through a TMDC monolayer arises due to valley Zeeman splitting as follows. The electric fields of the σ^{\pm} circularly polarized components of the incident linear polarization (assuming along \hat{y} for the sake of argument) $\vec{\mathcal{E}}(t) \propto \sin\left(\frac{E_l}{\hbar} t\right) \hat{y}$ interacting with the K^{\pm} valleys of the TMDC can be proportionately represented in the form of Jones vectors:

$$\vec{\mathcal{E}}_{K^{\pm}}(t) \propto \begin{pmatrix} \mp \cos\left(\frac{E_{\sigma^{\pm}}}{\hbar} t\right) \\ \sin\left(\frac{E_{\sigma^{\pm}}}{\hbar} t\right) \end{pmatrix} \quad (1)$$

The superimposed electric field of light after passing through the material is given as

$$\vec{\mathcal{E}}(t) \propto \sin\left(\frac{E_{\sigma^+} + E_{\sigma^-}}{2\hbar} t\right) \begin{pmatrix} \sin\left(\frac{\Delta E}{2\hbar} t\right) \\ \cos\left(\frac{\Delta E}{2\hbar} t\right) \end{pmatrix} \sim \sin\left(\frac{E_l}{\hbar} t\right) \left[\frac{\Delta E}{2\hbar} t \hat{x} + \hat{y} \right] \text{ for small } \Delta E \quad (2)$$

In the absence of an external magnetic field, $\Delta E = 0$. Therefore, $\vec{\mathcal{E}}(t) \propto \sin\left(\frac{E_l}{\hbar} t\right)$ with $E_l = E_{\sigma^+} = E_{\sigma^-}$ and there is no rotation of the electric field. However, in the presence of the magnetic field, the valley Zeeman splitting of the exciton ($\Delta E \neq 0$) results in a Faraday rotation of $\frac{\pi}{2} - \cot^{-1}\left(\frac{\Delta E}{2\hbar} t\right)$ towards \hat{x} .

We have included this additional detailed explanation on the microscopic origin of the Faraday rotation in the revised supporting information.

Referee comment. Authors claim: " However, in our present work, Landau quantization of the excitonic transitions does not happen in undoped or slightly doped 2D semiconductors until

under very high magnetic fields (>30 T)..". There are many theoretical works on the LL of monolayer TMDC under the application of magnetic field. Regarding to the magnitude of magnetic field, there is also the theoretical works on the LL in monolayer MoS₂ quantum Hall systems [PHYSICAL REVIEW B 90, 045427 (2014)], where magnetic field is around 5T. Why the authors can achieve the good FR at small magnetic field (1.4T)? And if small magnetic field applied here is their innovation point, the authors should present these in the introduction part.

Our response. We agree with the reviewer that there are many theoretical works on the LL of monolayers in TMDCs under the application of magnetic fields. The reviewer provides an example of one such work, where Landau levels in an electron gas in MoS₂ can be created around $B = 5$ T. The reviewer asks why we are able to achieve good FR at a smaller magnetic field in our experiment. We wish to emphasise that we are not working with a quantum Hall system here. Our monolayers are undoped (or unintentionally doped with very small doping), and our magnetic field is well below those required for creating LLs in a TMDC, as we have discussed in our original submission. The Faraday rotation which we achieve is not due to inter LL transitions. Instead, it is due to the shift of the conduction and valence bands at the K^{\pm} valleys in opposite directions, as explained above (Fig. R1). In that way, optical interband transitions at the K^{\pm} valleys undergo a valley Zeeman splitting, resulting in a large Faraday rotation. Following the referee's suggestions, we have highlighted this point in the introduction part of the revised manuscript at two locations on page 1 and 2 in red.

References.

1. Deilmann, T., Krüger, P. & Rohlfing, M. Ab Initio Studies of Exciton g Factors: Monolayer Transition Metal Dichalcogenides in Magnetic Fields. *Phys. Rev. Lett.* **124**, 226402 (2020).
2. Kormányos, A. *et al.* k·p theory for two-dimensional transition metal dichalcogenide semiconductors. *2D Mater.* **2**, 022001 (2015).
3. Wang, G. *et al.* Colloquium : Excitons in atomically thin transition metal dichalcogenides. *Rev. Mod. Phys.* **90**, 021001 (2018).
4. Mak, K. F., Xiao, D. & Shan, J. Light–valley interactions in 2D semiconductors. *Nat. Photonics* **12**, 451–460 (2018).
5. Koperski, M. *et al.* Orbital, spin and valley contributions to Zeeman splitting of excitonic resonances in MoSe₂, WSe₂ and WS₂ Monolayers. *2D Mater.* **6**, 015001 (2018).
6. Arora, A. Magneto-optics of layered two-dimensional semiconductors and heterostructures : Progress and prospects. *J. Appl. Phys.* **129**, 120902 (2021).
7. Arora, A. *et al.* Valley Zeeman Splitting and Valley Polarization of Neutral and Charged Excitons in Monolayer MoTe₂ at High Magnetic Fields. *Nano Lett.* **16**, 3624–3629 (2016).
8. Woźniak, T., Faria Junior, P. E., Seifert, G., Chaves, A. & Kunstmann, J. Exciton g factors of van der Waals heterostructures from first-principles calculations. *Phys. Rev. B* **101**, 1–29 (2020).
9. Mak, K. F., He, K., Shan, J. & Heinz, T. F. Control of valley polarization in monolayer MoS₂ by optical helicity. *Nat. Nanotechnol.* **7**, 494–498 (2012).
10. Zeng, H., Dai, J., Yao, W., Xiao, D. & Cui, X. Valley polarization in MoS₂ monolayers

- by optical pumping. *Nat. Nanotechnol.* **7**, 490–493 (2012).
11. Sallen, G. *et al.* Robust optical emission polarization in MoS₂ monolayers through selective valley excitation. *Phys. Rev. B* **86**, 081301 (2012).
 12. Srivastava, A. *et al.* Valley Zeeman effect in elementary optical excitations of monolayer WSe₂. *Nat. Phys.* **11**, 141–147 (2015).
 13. Xuan, F. & Quek, S. Y. Valley Zeeman effect and Landau levels in two-dimensional transition metal dichalcogenides. *Phys. Rev. Res.* **2**, 033256 (2020).
 14. Förste, J. *et al.* Exciton g-factors in monolayer and bilayer WSe₂ from experiment and theory. *Nat. Commun.* **11**, 4539 (2020).